# Reduced cellular binding affinity has profoundly different impacts on the spread of distinct poxviruses

**Erica B. Flores, Mee Y. Bartee, Eric Bartee** *

Division of Molecular Medicine, Department of Internal Medicine, University of New Mexico Health Sciences Center, Albuquerque, NM, United States of America

* ebartee@salud.unm.edu

**Data Availability Statement:** All relevant data are within the manuscript and its Supporting Information files.

**Funding:** EB R21AI123803 NIH (NIAID) https://www.niaid.nih.gov/ The funders had no role in

## Abstract

Poxviruses are large enveloped viruses that replicate exclusively in the cytoplasm. Like all viruses, their replication cycle begins with virion adsorption to the cell surface. Unlike most other viral families, however, no unique poxviral receptor has ever been identified. In the absence of a unique receptor, poxviruses are instead thought to adhere to the cell surface primarily through electrostatic interactions between the positively charged viral envelope proteins and the negatively charged sulfate groups on cellular glycosaminoglycans (GAGs). While these negatively charged GAGs are an integral part of all eukaryotic membranes, their specific expression and sulfation patterns differ between cell types. Critically, while poxviral binding has been extensively studied using virally centered genetic strategies, the impact of cell-intrinsic changes to GAG charge has never been examined. Here we show that loss of heparin sulfation, accomplished by deleting the enzyme N-Deacetylase and N-Sulfotransferase-1 (NDST1) which is essential for GAG sulfation, significantly reduces the binding affinity of both vaccinia and myxoma viruses to the cell surface. Strikingly, however, while this lowered binding affinity inhibits the subsequent spread of myxoma virus, it actually enhances the overall spread of vaccinia by generating more diffuse regions of infection. These data indicate that cell-intrinsic GAG sulfation plays a major role in poxviral infection, however, this role varies significantly between different members of the *poxviridae*.

## Introduction

Successful binding of a virion to a host cell is the first step in initiating any viral replication cycle. For many viral families, the impact of this step is well known. For members of the *poxviridae*, however, it remains somewhat poorly understood. In a large part, this is due to the fact that specific cellular receptors involved in poxviral adsorption have never been identified. Instead, poxviral binding is thought to be mediated by ubiquitously expressed, negatively charged molecules on the surface of host cells [1]. The best studied examples of this are the proteoglycans, which are comprised of glycosaminoglycan (GAG) side-chains covalently attached to any number of core proteins. The GAG portions of these proteoglycans are

study design, data collection and analysis, decision
to publish, or preparation of the manuscript. EB
R01CA194090 NIH (NCI) https://www.cancer.gov/
The funders had no role in study design, data
collection and analysis, decision to publish, or
preparation of the manuscript. EB RSG-17-047-01
American Cancer Society https://www.cancer.org
The funders had no role in study design, data
collection and analysis, decision to publish, or
preparation of the manuscript.

**Competing interests:** The authors have declared
that no competing interests exist.

characterized by repeating disaccharide units which are uniquely modified through a sequence of enzymatic reactions which regulate their ability to influence a wide variety of biological functions, including: proliferation, migration, cell adhesion, differentiation, and morphogenesis [2–5]. One of the key post-translational modifications undergone by GAGs is sulfation [6]. This modification occurs in several GAG families, including: heparin sulfate (HS), chondroitin sulfate and dermatan sulfate, and imparts a large negative charge to GAGs on the cell surface. While GAG sulfation is associated with a variety of critical cellular functions, a number of viruses, including: dengue virus, herpes simplex virus, hepatitis B virus, respiratory syncytial virus, and members of the *poxviridae* [7–10], have also coopted the negative charge it produces as a means to enhance viral adsorption. This interaction is thought to be mediated by positively charged viral membrane or capsid proteins engaging in electrostatic interactions with the negative charge associated with sulfated GAG moieties.

Specifically within the *poxviridae*, several viral envelope proteins from vaccinia virus (VACV) have been identified that bind to sulfated cell surface GAGs including A27L and H3L which bind specifically to HS and D8L which binds to chondroitin sulfate [11–13]. Genetic loss of these viral genes results in lower infectivity, reduced production of infectious progeny, and smaller plaque sizes [13] while pharmacological blockade of the electrostatic virion:GAG interaction using soluble HS can largely prevent poxviral infection [11, 14]. Critically, however, no studies have examined the impact of GAG sulfation on poxviral replication from a cell-intrinsic view. In order to understand how this sulfation impacts the poxviral replication cycle, we therefore established sulfation deficient cells and examined how loss of this GAG modification impacted the replication cycles of two model poxviruses, the classic orthopoxvirus VACV as well as the leporipoxvirus myxoma (MYXV).

## Materials and methods

### Reagents and viruses

MYXV (strain Lausanne) expressing green fluorescent protein (GFP) under regulation of the consensus poxviral synthetic early/late promoter has been previously described [15]. VACV (strain WR) expressing GFP was a kind gift from Dr. Paula Traktman at the Medical University of South Carolina. Unless otherwise noted, viral infections were carried out by inoculating cells with the indicated viral multiplicity of infection (MOI) for one hour and subsequently replacing inoculum with fresh growth media. Direct labeling of MYXV and VACV virions with Cy5 was done as previously described [14]. BSC40 cells (Cat# CRL-2761) were purchased from American Type Culture Collection (ATCC, Manassas, VA, USA). B16/F10 cells were a kind gift form Dr. Chrystal Paulos at the Medical University of South Carolina. All cells were maintained in Dulbecco's Modified Eagle Medium (DMEM, Mediatech, Inc., Manassas, VA, USA) supplemented with 10% fetal bovine serum (VWR, Radnor, PA, USA) and 100 U/ml penicillin-streptomycin (Mediatech, Inc., Manassas, VA, USA). The following antibodies were used in these studies. For flow cytometry: sulfated heparin (clone F58-10E4, AMS Biotechnology, Cambridge, MA, USA). Direct conjugation of the 10E4 antibody to PE/Cy7 was done using the PE/Cy7 conjugation kit (ab102903, Abcam, Cambridge, UK) according to manufacturer recommendations. For western blot: NDST1 (clone E-9) and actin (clone I-19) (Santa Cruz Biotechnology, Inc., Dallas, TX, USA).

### Generation of NDST deficient cells

N-Deacetylase and N-Sulfotransferase-1 (NDST1) deficient B16/F10 cell lines were generated using the CRISPR/Cas9 system (Genscript, Piscataway, NJ, USA) as previously described [16]. In short, B16/F10 cells were transfected with the plasmid pSpCas9BB-24-2A-Puro containing

a NDST1 specific gRNA (seq AAGCCACGGCGGTACCGGGC). 48 hours after transfection, cells were transferred to media containing 1μg/ml puromycin for one week to select for transfected cells. Cells were then removed from selective media and single cells were expanded as individual clonal lines. For these experiments, two clones were used, referred to as NDST1[-/-] #1 and NDST1[-/-] #2. Control B16/F10 cells expressing NDST1 (NDST1[+]) treated with a scrambled gRNA (seq GCGAGGTCTTCGGCTCCGCG) have been described previously [16].

## Quantitative PCR (qPCR)

mRNA was extracted from cells using the RNEasy kit (Qiagen, Hilden, Germany) and cDNA was synthesized using the Superscript IV VILO™ master mix kit (ThermoFisher Scientific, Waltham, MA, USA) according to manufacturer's recommendations. Synthesized cDNA was them mixed with the PowerUp™ SYBR™ Green master mix kit (ThermoFisher Scientific, Waltham, MA, USA) and abundance of target transcripts detected on a CFX96™ Real-Time System (Bio-Rad, Hercules, CA, USA). Data was analyzed with the accompanying Bio-Rad CFX Manager™ software. Viral genomes were extracted using the Quick-DNA™ Miniprep Plus Kit and were detected by the same methods mentioned previously. Amplified products were analyzed by gel electrophoresis to ensure specificity. PCR primers used in this study are shown in Table 1.

## Flow cytometry analysis

Cells were harvested and washed several times with PBS to remove residual media before incubation with virus/antibody for 30 min at 4°C. After additional washes, cells were resuspended in 2% paraformaldehyde (PFA) and samples were analyzed on a BD FacsVerse cytometer (BD Biosciences, San Jose, CA, USA).

## Analysis of viral replication cycle

Direct binding of virus to the cell surface [14], intracellular single-step growth curves [17], and initiation of viral infection [18] were analyzed as previously described. Viral foci/plaque size was determined by drawing a region of interest around the outermost GFP[+] cells within a specific infection and subsequently calculating the area of that region using ImageJ software. Examples of regions drawn to analyze foci/plaque area are included as S1 Fig. To account for variation within measuring, analysis was done on numerous foci/plaques across several independent experiments. Viral foci/plaque formation under methyl cellulose was analyzed by adding 2ml of DMEM containing methyl cellulose directly after removal of viral inoculum.

## Results

### Loss of NDST1 prevents sulfation of cell surface heparin

There are four NDST enzymes involved in heparin sulfation [19], however, removal of NDST1 alone has been shown to be sufficient to completely prevent GAG sulfation [20]. To investigate

**Table 1. List of primers used in this study.**

| | qPCR Primer Sequences | |
| --- | --- | --- |
| | Forward | Reverse |
| **NDST-1** | CCCACTGGTGCTGGTATTT | TGCAATCTCTGTCCGGTATTT |
| MO55 | ACGGACATCTCTCCCAGACA | TGCACGTCGGGTTTATTTGC |
| MO85 | ACGGCATTTAACAACCAGCG | CATCGCACGATCTCGGAGTA |

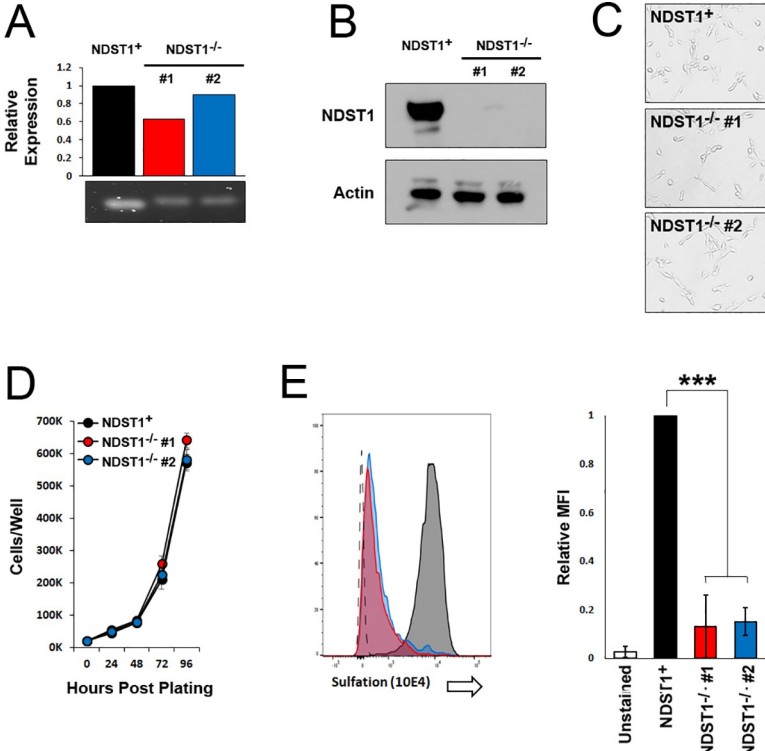

**Fig 1. Generation of cells deficient in sulfation of cell surface heparin chains.** (A) PCR analysis of NDST1 mRNA expression in NDST1+ and NDST1 deficient cell lines. Bar graph shows quantitative data from real-time PCR analysis. Qualitative agarose gel is shown for visual purposes. (B) Western Blot analysis was performed using the indicated antibodies to detect expression of NDST1. Expression of actin is shown as a loading control. Data is representative of two independent experiments. (C) Phase contrast images of NDST1+ and deficient cells depicting general overall cellular morphology. (D) Cell growth of both NDST1+ and deficient cells measured over a 96-hour period. Data is representative of two independent experiments. (E) Sulfation of cell surface heparin measured via flow cytometry utilizing an antibody that recognizes the sulfated 10E4 epitope on HS chains. The mean fluorescent intensity (MFI) values were then normalized to NDST+ values to show the relative change in the NDST-/- cell lines. Data is representative of three independent experiments. Statistical significance was determined using Students T-Test. *** = p<0.001.

the impact of cell-intrinsic GAG sulfation on poxviral replication we therefore generated cells unable to express NDST1 due to CRISPR/Cas9 genomic editing. B16/F10 cells (which express endogenous NDST1 and NDST2, but not NDST3 or NDSTD4 S2 Fig) were transfected with a plasmid expressing the CRISPR/Cas9 enzymes as well as a gRNA targeting the NDST1 open reading frame. Following puromycin selection, two putative clonal NDST1-/- cell lines (NDST1-/- #1 and NDST1-/- #2) were grown from individual, isolated cells. A control NDST1 expressing clonal line (NDST1+) was previously generated following identical treatment of cells with a plasmid expressing CRISPR/Cas9 and a scrambled gRNA [16]. Consistent with the mechanisms of CRIPSR/Cas9 editing, qPCR analysis performed on cDNA synthesized from NDST1+ cells or either NDST1-/- cell line identified similar levels of NDST1 transcript (Fig 1A). In contrast, western blot analysis showed a complete loss of NDST1 protein expression in NDST1-/- #1 and a near complete loss in NDST1-/- #2 (Fig 1B). This loss of NDST1 protein expression did not obviously affect the overall morphology of the cells (Fig 1C) and did not change cellular growth rates over a 96-hour period (Fig 1D). Both NDST1 deficient cell lines, however, displayed a drastic decrease in staining with the commercial 10E4 antibody, which specifically recognizes a sulfated epitope on heparin sulfate chains (Fig 1E).

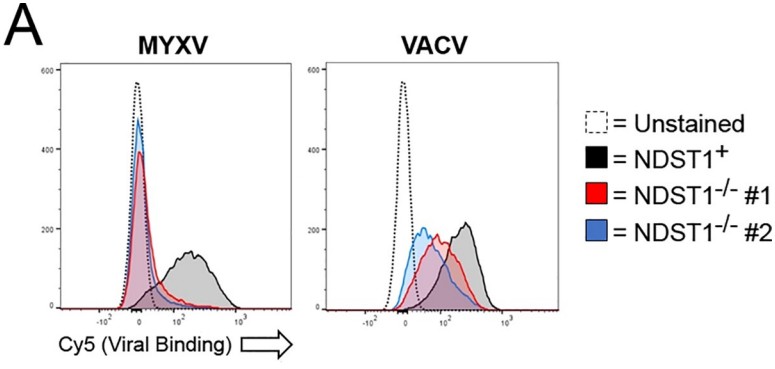

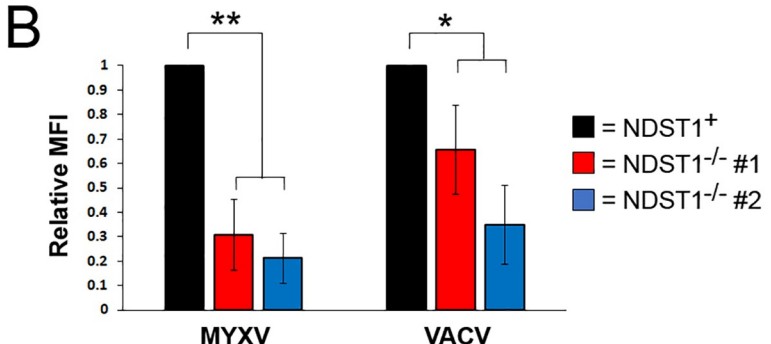

**Fig 2. Loss of sulfation reduces binding of both MYXV and VACV virions.** (A) Histograms depicting binding of fluorescent viral virions to the cell surface. Cells were incubated on ice with Cy-5 labeled MYXV or VACV at 4 particles per cell and washed with cold PBS to remove residual virus. Cells were resuspended in 2% PFA and levels of Cy-5-labeled virus bound to the cell membrane was measured via flow cytometry. Data shown is representative of two independent experiments each done in triplicate. (B) Relative MFI values of Cy-5 (viral binding). MFI values were normalized to NDST[+] to determine the relative change in NDST[-/-] cell lines. Statistical MFI calculations are a summation of the two independent experiments each conducted in triplicate. Statistical significance was determined using non-parametric Mann-Whitney test. ** = $p < 0.01$, * = $p < 0.05$.

## NDST1[-/-] cells show a reduced affinity for poxviral binding

In order to determine whether the loss of heparin sulfation affects the poxviral replication cycle, we first assayed the direct binding of two model poxviruses, the classical orthopoxvirus VACV as well as the leporipoxvirus MYXV, to both NDST1[+] and NDST1[-/-] cells. Cells were incubated on ice with virions directly labeled with fluorescent Cy5 dye. Cells were then washed to remove unbound virions and the amount of virus bound to the cell surface was analyzed by detecting Cy5 fluorescence using flow cytometry. We observed a clear reduction in Cy5 fluorescence in both NDST1[-/-] cell lines compared to that observed on NDST1[+] cells (Fig 2A). This reduction was seen for both VACV and MYXV, however, it was more pronounced for MYXV (which displayed a 4.0±0.7-fold decrease in Cy5 mean fluorescent intensity) then for VACV (which displayed a 2.0±0.7-fold decrease). No significant differences in fluorescent intensities were observed between the two NDST1[-/-] cell lines (Fig 2B).

## NDST1[-/-] cells display a delay in intracellular poxviral replication

Since our previous experiments suggested that loss of heparin sulfation significantly reduced the affinity of viral binding, we next wanted to determine whether this reduction correlated to lower subsequent infection. Both NDST1[+] and NDST1[-/-] cells were infected with MYXV and

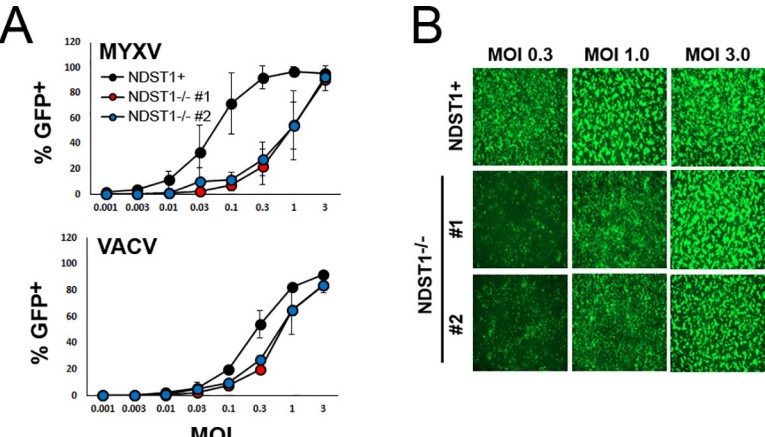

**Fig 3. Loss of sulfation reduces initial infection of both MYXV and VACV.** (A) Number of GFP⁺ cells found 24 hours after infection with either MYXV or VACV at multiple MOI's. Data shown is the summation of two independent experiments each conducted in triplicate. (B) Fluorescent images of cells infected for 24 hours with MYXV at an MOI of 1, 3, and 9 shown as an example.

VACV at MOIs ranging from 0.0001 to 3. The initiation of viral infection was then assayed by determining the number of GFP⁺ cells found in each culture 24 hours post infection using flow cytometry. The results indicated that infection of NDST1 deficient cells resulted in a lower percentage of GFP⁺ cells over a wide range of MOIs (Fig 3A and 3B). As with binding affinity, this reduction was observed for both viruses, however, it was again more pronounced for MYXV than for VACV. Interestingly, infection with either virus at a high MOI (over 3) resulted in a near 100% rate of infection in both NDST1⁺ and NDST1⁻/⁻ cells. Taken together, these data suggest that loss of heparin sulfation reduces initial poxviral infection, however, this reduction can be overcome with high concentrations of virus.

To better resolve how the loss of heparin sulfation impacts the early steps of poxviral replication, we next performed a time course assaying expression of GFP as a surrogate marker for viral entry and/or early gene expression. NDST1⁺ and NDST1⁻/⁻ cells were infected at an MOI of 10 with either MYXV or VACV and the expression of GFP was measured every 2 hours via flow cytometry. GFP expression was also analyzed at late time points including 24 hours as well as at 36 and 48 hours for VACV. During MYXV infection, GFP expression became detectable in the majority of NDST1⁺ cells between 2 and 6 hours post infection. In contrast, 6 hours after infection of NDST1 deficient cells, GFP could be detected in only ~20% of cells and detection in the majority of cells was not observed until 24 hours post infection. A similar trend was observed following VACV infection, although the overall kinetics of viral early gene expression were significantly delayed for VACV compared to MYXV. Indeed, overall VACV infection in this experiment was inefficient compared to other previous infections (Fig 3) which is likely due to minor experimental variation. In VACV infected cells, GFP was not detectable in most NDST1⁺ cells until 12–36 hours post infection and in most NDST1⁻/⁻ cells until 24–48 hours post infection (Fig 4A). These kinetics are slightly delayed compared to previous results (Fig 3A) which we attribute to slight biological differences across experiments. To assess whether this delay in early gene expression hindered subsequent viral genome synthesis, both NDST1⁺ and NDST1⁻/⁻ cells were infected with MYXV of VACV at an MOI 10 and samples collected every two hours for 12 hours. Total DNA was then extracted and the abundance of viral genomes was determined by using qPCR to detect either the MYXV- M085 or VACV-L1R genomic loci (Fig 4B). Consistent with the kinetics of poxviral replication, genomic viral DNA

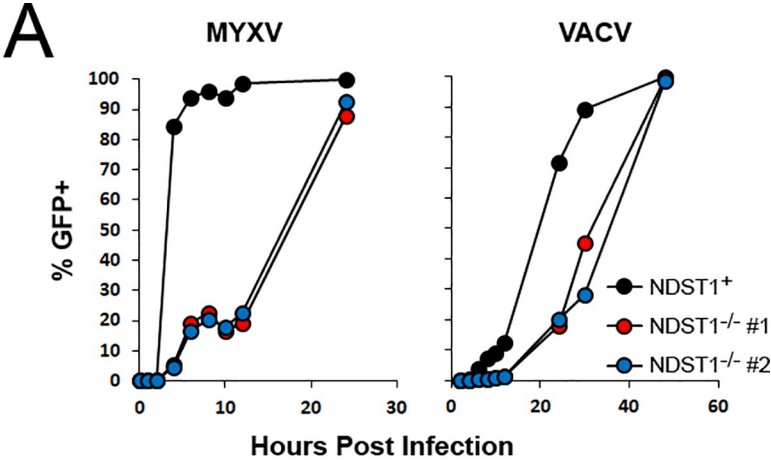

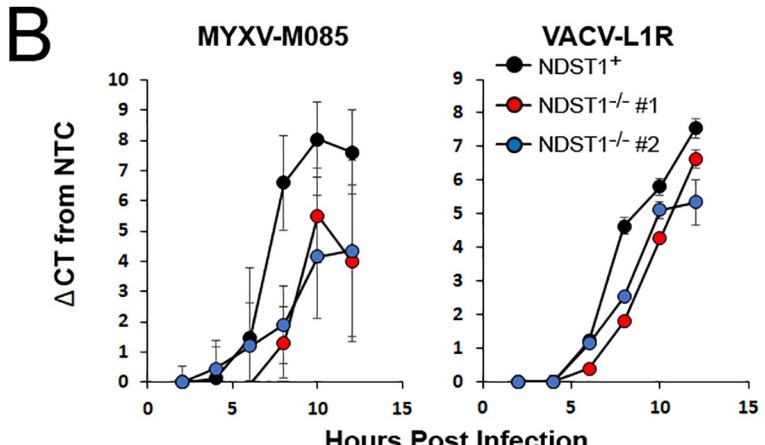

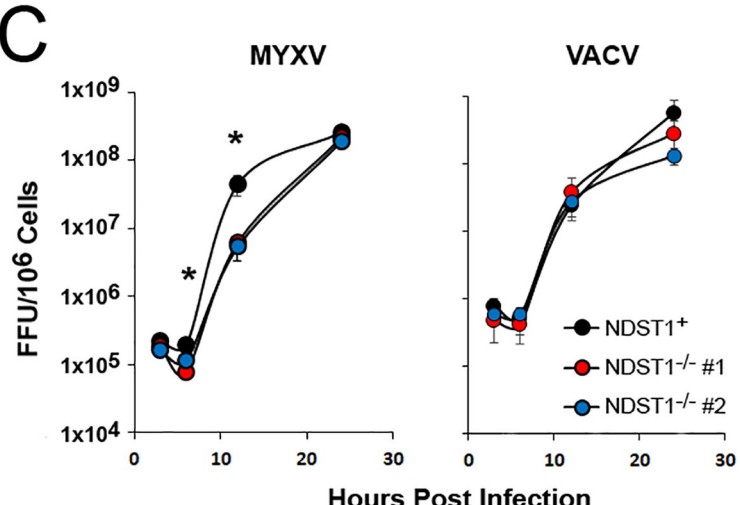

**Fig 4. Loss of sulfation delays intracellular poxviral replication.** (A) Kinetic analysis of GFP expression after initial infection of NDST1+ or deficient cells. NDST1+ or deficient B16/F10 cells were infected with either MYXV or VACV and GFP expression measured every two hours using flow cytometry. Data presented is representative of two independent experiments. (B) Abundance of MYXV or VACV genomes measured using qPCR. Data presented is representative of three independent experiments for both viruses each analyzed in triplicate and is presented as the change in cycle threshold value (ΔCT) from 2 hour baseline. (C) Production of new infection viral progeny in either

NDST1$^+$ or deficient cells measured using standard single step growth curve analysis. Data presented is a summation of two (VACV) or three (MYXV) independent experiments each conducted in triplicate. Statistical significance was determined using Students T-Test (* = p<0.05).

could not be reliably detected in NDST1$^+$ cells until 6 hours post infection, at which point we observed a significant increase in signal for both MYXV and VACV genomes. In NDST1$^{-/-}$ cells, a similar increase in signal was observed for both viruses, however, this increase was of lesser magnitude and occurred at slightly later time points suggesting both reduced and delayed genome synthesis. Identical results were also obtained for both viruses using a second genomic loci (M055 for MYXV and D10R for VACV–data not shown). Finally, to determine whether the apparent delay in genome synthesis would impair the production of new infectious progeny, we quantified viral assembly using a single step growth curve. NDST1$^+$ or deficient cells were infected with either MYXV or VACV at an MOI of 10. Following infection, cells were harvested at the indicated time points and the presence of infectious virus determined using standard foci/plaque forming assays. While the results clearly indicated productive viral replication in both NDST1$^+$ and deficient cells, significantly fewer infectious MYXV virions were observed in NDST1$^{-/-}$ cells at both 6 and 12 hours post infection (Fig 4C). The number of infectious MYXV particles normalized by 24 hours post infection and no significant differences in the assembly of VACV were detected at any time point (Fig 4C Right).

## Reduced binding affinity has profoundly different effects on MYXV and VACV spread

Our previous experiments established that loss of NDST1 compromised the overall intracellular replication of both MYXV and VACV. In order to determine whether this compromise would impact viral spread, we next asked how the presence or absence of NDST1 influenced the foci/plaque size of each virus. NDST1$^+$ or deficient cells were infected with either MYXV or VACV at an MOI of 0.001. Images of individual GFP$^+$ foci/plaques were taken at 24, 48, and 72 hours post infection and the area of each infected region was quantified using imaging software (Fig 5A and 5B). Consistent with its compromised replication, GFP$^+$ foci resulting from MYXV infection were ~50% smaller in both NDST1$^{-/-}$ cell lines compared to those seen in NDST1$^+$ cells (p<0.005 at 72 hours). In striking contrast, however, while the intracellular replication of VACV was also clearly compromised by the loss of NDST1, we found that the spread of VACV in both NDST1$^{-/-}$ cell lines was significantly increased compared to that seen in the NDST1$^+$ cells at both 48 and 72 hours post infection (2.0±0.1-fold larger in NDST1$^{-/-}$ cells, p<0.005) (Fig 5B). This unexpected result led us to look more closely at the composition of the VACV plaques found in each cell type (Fig 5C). We observed that VACV plaques formed in NDST1$^+$ cells contained a relatively tight center of closely packed infected cells surrounded by a small ring of more diffuse infection. In contrast, plaques formed in NDST1 deficient cells largely lacked the tightly packed center core and instead contained only a large group of diffusely infected cells. To determine if this distribution of infection was solely responsible for the increased plaque sizes seen in NDST1$^{-/-}$ cells, we further quantitated the total amount of infection within individual plaques by analyzing overall GFP expression. For MYXV, this analysis showed that the total GFP content of individual foci was significantly reduced in NDST1$^{-/-}$ cells corresponding to their decreased area. In contrast, the total GFP content of VACV plaques was actually identical in both NDST1$^+$ and deficient cells suggesting that the observed change in plaque size was due to an altered distribution of infection and not an actual increase in total virus present (Fig 5D).

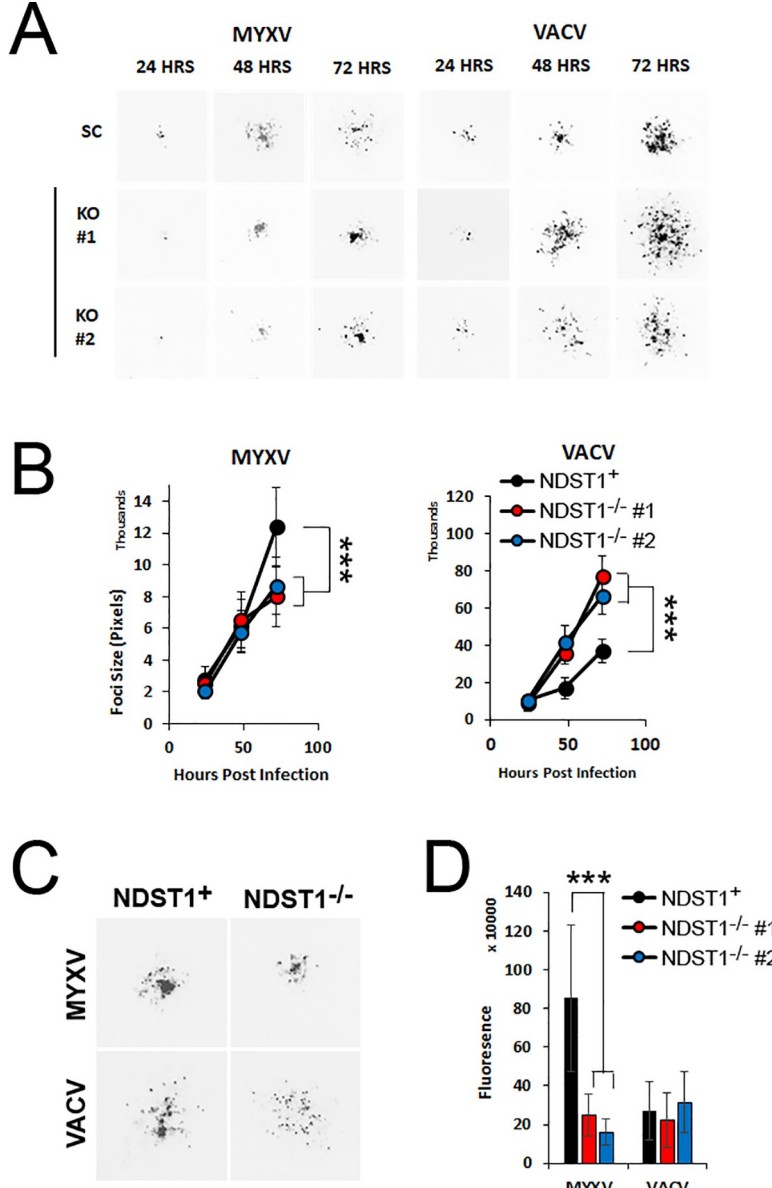

**Fig 5. Binding affinity has inverse impacts on the spread of different poxviruses.** (A) Images of individual GFP+ foci formed in either NDST1+ or deficient cells taken 24, 48, or 72 hours post infection with the indicated virus. (B) Quantitation of individual foci size. Data shown is representative of four independent experiments, where a total of >60 foci per cell/virus type were measured. (C) Up-close image of foci showing the different concentration of infected cells in the core. (D) Quantitation of total GFP signal in individual foci from experiments above. Data shown represents average GFP expression from >15 foci measured across three experiments. Statistical significance was determined using Students T-Test (*** = p<0.001).

One of the major differences between orthopoxviruses and MYXV is that MYXV is largely unable to produce infectious extracellular enveloped virus (EEV) [1] due to its lack of an *f11l* homologue [21]. To test whether the differential impact of GAG sulfation on viral spread which we observed in our previous experiments might be related to the presence/absence of infectious EEV we therefore used methyl cellulose to inhibit the release of EEV and subsequently measured foci/plaque size of both VACV and MYXV infections in NDST1+ or

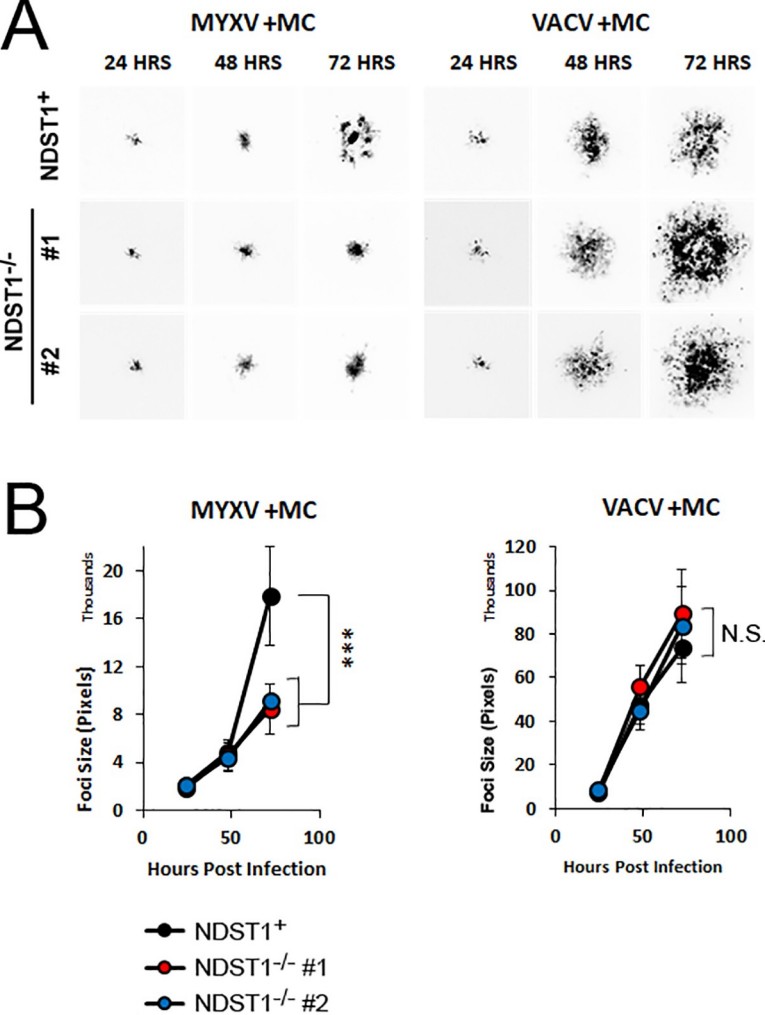

**Fig 6. Increased spread seen during low affinity VACV infections is mediated by secreted virions.** (A) Images of individual GFP+ foci formed in either NDST1+ or deficient cells covered with methyl cellulose. Images were taken 24, 48, or 72 hours post infection with the indicated virus. (B) Quantitation of individual foci size. Data is representative from at least three independent experiments. Statistical significance was determined using Students T-Test. *** = p<0.001.

deficient cells. Consistent with our results in the absence of methyl cellulose, under EEV-restricting conditions MYXV formed significantly smaller plaques in NDST1-/- cells than in NDST1+ cells. In contrast, in the presence of methyl cellulose, plaques formed by VACV were identical in size in both NDST1+ and deficient cells (Fig 6A and 6B) suggesting that the large plaque phenotype observed in our previous experiments (Fig 5) was the result of viral spread through EEV.

## Discussion

This major aim of this study was to determine how the progression of poxviral infection can be affected by cell-intrinsic factors which influence virion binding affinity. To accomplish this, we generated a cell line which is unable to add sulfates onto cell surface heparin proteoglycans due to the loss of the enzyme NDST1, which is essential for heparin sulfation. Interestingly, while GAG sulfation has been shown to play a major role in a wide array of cellular processes

[22, 23], loss of NDST1 did not appear to grossly alter either the morphology or growth properties of B16/F10 cells *in vitro* (Fig 1) suggesting that these cells represent a feasible model to study poxviral infection under either high affinity (NDST1$^+$) or low affinity (NDST1$^{-/-}$) conditions.

Consistent with a role for negatively charged GAGs in poxviral binding, the loss of NDST1 correlated with a significant reduction in virion binding for both MXYV and VACV. Interestingly, this reduction was much more dramatic for MYXV than for VACV (Fig 2) which is similar to results from previous studies demonstrating that these two viruses display differential binding specificities for certain cell types [24]. This reduction in binding affinity resulted in reduced rates of infection in NDST1$^{-/-}$ cells across a range of MOIs. This decreased infection, however, could be overcome with high concentrations of virus (MOI's > 3). This could be due to incomplete loss of heparin sulfation following NDST1 removal or to a HS-independent binding mechanism. In support of the second hypothesis, a recent study found that VACV infection was only moderately reduced when HS was removed by heparinase digestion, as opposed to near complete inhibition following treatment of virions with soluble heparin [25]. The authors of this work suggested that the difference might be due to incomplete heparinase digestion, however, our work suggests it is more likely due to an inherent property of VACV binding. In general, both this work and our current results suggest that VACV is effectively able to bind cells in the absence of sulfated heparin, while MYXV binding is much more HS dependent. This could be due to a VACV having an inherent affinity for other sulfated GAGs, such as chondroitin sulfate, however our work does not explore this possibility.

Even at high MOI's, where the defects in early viral infection appeared to be overcome, lack of GAG sulfation was still associated with delayed early gene expression, genome synthesis, and assembly of new progeny virions (Fig 4A–4C). There are two possible explanations for these results. First is that progression through the viral replication cycle is dose dependent. This is supported by the general trend of poxviruses to display particle: PFU ratios above 10 without any obvious presence of defective interfering particles [26]. Alternatively, the presence of cell surface HS could be involved in directing post-binding steps of the poxviral replication cycle, most likely the early post-binding steps such as viral entry and/or uncoating. The observed delays in subsequent steps, such as genome replication and assembly, would then likely be the result of delayed early replication without requiring a direct involvement of HS in these processes. This model is partially supported by the delayed early gene expression, as measured by GFP, seen in Fig 4A, however, more direct measures of viral entry and/or uncoating would likely help clarify this point.

While the previous results could likely be anticipated based on existing studies, our data also show that the reduced binding affinity caused by loss of heparin sulfation has completely opposite effects on the spread of the two model poxviruses studied. In the case of MYXV, we found spread to be greater in NDST1$^+$ than NDST1$^{-/-}$ cells and this trend remained consistent with the addition of methyl cellulose. In sharp contrast, we found that the spread of VACV was actually greater in NDST1$^{-/-}$ cells and that this difference was eliminated by the presence of methyl cellulose (Figs 5 and 6). We hypothesize that the differential impact of heparin sulfation on the foci/plaque size of MYXV and VACV may be attributed to the mechanisms of how each virus spreads. It is well established that most poxviruses produce two forms of infectious particles: intracellular mature virion (IMV) as well as EEV, which is IMV wrapped in an at least one additional cell membrane layer. Each of these forms differs in their physical and chemical structure and displays a distinct set of proteins on their surface, [27]. These differences can translate to distinct binding properties, with EEV having a greater affinity for interactions with highly sulfated HS than IMV [25]. VACV produces typical amounts of EEV which are involved in viral transmission. In contrast, MYXV produces little to no EEV,

although it can produce extracellular, cell associated enveloped virus (CEV). In the case of MYXV, this strictly limits spread to direct cell to cell contact. Under low affinity binding conditions, this likely reduces the efficacy of spread during every round of viral replication resulting in an overall reduced foci size. In contrast, while VACV can also spread through direct cell to cell contact, it produces a more substantial amount of EEV which can transmit virus through the extracellular space to non-neighboring cells. Under high affinity conditions, these extracellular particles are likely to bind rapidly to neighboring cells since they readily adsorb to the cell surface. In contrast, under low affinity conditions, each cell is less susceptible to acute viral adsorption allowing EEV particles to potentially bypass neighboring cells and instead continue trafficking to sites further from their initial release. This model explains why VACV generates larger plaques in NDST1$^{-/-}$ deficient settings without actually infecting a greater number of cells (Fig 5). Interestingly, this effect is similar to the viral repulsion effect which was previously proposed as a mechanism to enhance the spread of VACV by preventing superinfection [28]. In the previous work, however, the mechanism of repulsion was expression of two viral proteins (A33 and A36) while in our work the mechanism is cell-intrinsic. This model also explains why VACV foci formed in the presence of methyl cellulose appear to display increased overall density compared to foci formed in the absence of methyl cellulose (compare Figs 5 and 6). Numerous studies have shown that the particle to PFU ratio for poxviruses ranges from 10–100:1. Since no obvious defective poxviral particles have been identified, this suggests that the number of particles required to induce a productive infection within a single cell is greater than one. We hypothesize that methyl cellulose traps viral particles increasing the localized concentration, thus enhancing the rates of infection within a limited area.

An interesting correlate of this work is the possibility that proteins within the poxviral virion itself might contain sulfated GAG's that play a role in the poxviral life cycle. Numerous cellular proteins have been shown to be incorporated into the VACV and MYXV virions [29, 30] and several of these are likely candidates for GAG modification. Given the wide-spread impact of GAGs on numerous biological processes, such modifications could play any number of roles in poxviral infection. It is interesting to note, however, that poxviral particle binding is generally thought to be dependent on the virion carrying a large positive charge. Sulfation of virion GAG's would likely disrupt this charge thus possibly acting as a negative regulatory mechanism. Future studies into the potential role of poxviral protein sulfation might therefore yield interesting results.

Overall, these results highlight the importance of understanding poxviral binding and infection as a two-way interaction between the virus and host cell. While many studies have utilized genetically modified viruses or the addition of exogenous compounds, such as soluble HS, this research focused on how cell intrinsic factors influence poxviral biology. These findings highlight an unanticipated result of reducing poxviral binding affinity which could have a significant impact on how these viruses impact either human disease or poxviral mediated oncolytic virotherapy.

## Supporting information

**S1 Fig. Examples of foci/plaque quantitation.** Shown are regions drawn around individual VACV plaques 48 hour after infection of the indicated cells (region shown as white circle) given as examples of how foci/plaque size was determined. Foci area calculated for each region using ImageJ is shown below image. Note that these specific images/regions are presented only as after the fact examples of how regions were drawn. The specific visual images of the regions drawn for data acquisition were not saved in ImageJ and therefore only the calculated areas for

each foci/plaque remain.
(DOCX)

**S2 Fig. Expression of NDST1-4 in B16/F10 cells.** mRNA was extracted from untreated wild-type B16/F10 cells and used to synthesizes cDNA. Two different primer sets (corresponding to each previously reported NDST enzyme—NDST1, NDST2, NDST3, and NDST4) were then used to attempt to amplify regions of each gene from the cDNA. Successful PCR amplification was observed with both primer sets corresponding to NDST1 and NDST2. No specific PCR products were observed in either primer set against NDST3 or NDST4. Note that due to low technical quality the image shown has been enhanced for both brightness and contrast as well as cropped to remove irrelevant lanes on the right side.
(DOCX)

**S3 Fig. Original scan of Actin western blot used in Fig 1B.** Original scan of NDST western blot used in Fig 1B.
(PDF)

## Acknowledgments

We would like to thank Dr. Paula Traktman for valuable discussion about this project.

## Author Contributions

**Conceptualization:** Eric Bartee.

**Data curation:** Erica B. Flores, Eric Bartee.

**Formal analysis:** Erica B. Flores, Eric Bartee.

**Funding acquisition:** Eric Bartee.

**Investigation:** Erica B. Flores, Eric Bartee.

**Methodology:** Erica B. Flores, Mee Y. Bartee, Eric Bartee.

**Project administration:** Eric Bartee.

**Resources:** Mee Y. Bartee.

**Supervision:** Eric Bartee.

**Writing – original draft:** Erica B. Flores.

**Writing – review & editing:** Erica B. Flores, Mee Y. Bartee, Eric Bartee.

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
