## [Decision Letter · Decision Letter 0]

21 Jan 2020

PONE-D-19-34908

Reduced cellular binding affinity has profoundly different impacts on the spread of distinct poxviruses.

PLOS ONE

Dear Associate Professor Bartee,

Thank you for submitting your manuscript to PLOS ONE. After careful consideration, we feel that it has merit but does not fully meet PLOS ONE’s publication criteria as it currently stands. Therefore, we invite you to submit a revised version of the manuscript that addresses the points raised during the review process.

Please see the enclosed editor's comments below to guide your in preparing the resubmission.

We would appreciate receiving your revised manuscript by Mar 06 2020 11:59PM. To enhance the reproducibility of your results, we recommend that if applicable you deposit your laboratory protocols in protocols.io, where a protocol can be assigned its own identifier (DOI) such that it can be cited independently in the future. For instructions see: http://journals.plos.org/plosone/s/submission-guidelines#loc-laboratory-protocols

We look forward to receiving your revised manuscript.

Kind regards,

Luis M Schang, MV. Ph.D.

Academic Editor

PLOS ONE

3. Please include your tables as part of your main manuscript and remove the individual files. Please note that supplementary tables (should remain/ be uploaded) as separate "supporting information" files

Additional Editor Comments (if provided):

As you can see in the enclosed reviews, both reviewers have favorable opinions of the submitted manuscript. However, both of them also noted that the q-PCR experiments should be repeated, and this editor concurs with that opinion. Please make sure to also address all the editorial comments from both reviewers, too, as these changes will add clarity to the manuscript. Looking forward to receiving the revised manuscript with the q-PCR experiments repeated.

Reviewers' comments:

Reviewer's Responses to Questions

**Comments to the Author**

1. Is the manuscript technically sound, and do the data support the conclusions?

Reviewer #1: Yes

Reviewer #2: Partly

2. Has the statistical analysis been performed appropriately and rigorously? 

Reviewer #1: Yes

Reviewer #2: Yes

3. Have the authors made all data underlying the findings in their manuscript fully available?

Reviewer #1: Yes

Reviewer #2: Yes

4. Is the manuscript presented in an intelligible fashion and written in standard English?

Reviewer #1: Yes

Reviewer #2: Yes

5. Review Comments to the Author

Reviewer #1: The authors have used CRISPR/Cas9 technology to inactivate the sulfation enzyme NDST1 in mouse B16/F10 cells and then tested what effect this has on the growth and plaquing properties of VAC and MYX viruses. Mutating the gene delays establishing an infection but this can be compensated by using higher MOIs. They show that the two viruses differ somewhat in that the Leporipoxvirus forms relatively smaller plaques on k/o cells while the Orthopoxvirus forms larger and more diffuse plaques on NDST-/- cells versus NDST+ cells. The authors suggest that the differences between the two viruses might be due to somewhat different roles that EEV play in spreading VAC versus MYX from cell-to-cell.

On the whole I though that the work was well done and interpreted appropriately. The paper offers a useful counterpart to earlier studies concerning the roles of virus proteins. I did have a few comments:

1) Much of the discussion hinges on the measurement of "plaque size", but how this was determined by the authors is not clearly explained. This is actually quite tricky judging by the images shown in Fig5C or 6A, since there aren't clear margins to the plaques. Is it a measure of the diameter from the outermost of the GFP+ pixels or is it perhaps the area enclosing some % of the GFP+ signal? Some clarity here would be helpful.

2) The qPCR data in Fig 4B are odd. Given the discrepant measurements at 10 and 12 hr (in cell line #1 and 2), it's very difficult to come to any conclusions regarding whether DNA synthesis is affected by this mutation. This experiment should be repeated. Ideally one would like to know whether the replication kinetics are any different once the barrier to infection has been overcome (i.e post entry). That means identifying newly infected cells and measuring DNA synthesis in just those cells. Is there any way to determining this? For example, could one measure the rate of growth of DAPI strained virus factories once they are first detected?

3) Lines 251-62. The paragraph surrounding the role of EEV is confusing. The authors lead with a statement that MYX produce much less EEV than VAC, then use methyl cellulose to "inhibit release of EEV", see no effect of methyl cellulose on VAC plating in either cell type, and conclude that EEV play a major role in determining VAC spread. I can't follow the logic, there seems to be a non sequitur or circular reasoning somewhere. As an aside, VAC strain WR doesn't produce much EEV, the effect seen here might be more striking if strain IHD-W was used.

Lastly, one is still left uncertain whether this mutation is affecting binding alone or binding plus entry via the entry fusion complex. Fig 2 doesn't really help clarify this point and it's only briefly discussed in lines 296 - 300. Have the authors tried to measure a time lag between binding (for example on ice) and uncoating?

Minor points.

1) Abbreviations need to be explained (e.g. NTC)

2) The green plaque images (e.g. Fig 5A) would be better presented as black on white as in Fig 5C.

3) There's an NDST2 I gather, would it still be functional?

4) Can one call NDST1 "essential" as the authors do in the abstract? The k/o cells seem to grow fine.

As an experiment for another day I would be curious to know if virus grown on k/o cells have any subsequent problems infecting normal cells? Presumably the progeny aren't sulfated properly.

Reviewer #2: In this manuscript, Flores et al. evaluate the role of heparin sulfation in poxvirus entry, infection and spread. The authors used CRISPR to delete the sulfation enzyme NDST1 in murine B16/F10 cells. They show that deletion of NDST1 reduces binding and infection of two different poxviruses, MYXV and VACV, although the effect could be overcome by a high MOI. Notably, deletion of NDST1 inhibited spread of MYXV, but enhanced spread of VACV. Overall, the study is nicely designed, with experimental data that are appropriately interpreted and support the conclusions. However, a few additional experiments and explanations/clarifications would strengthen the manuscript.

Specific comments:

1. Can the authors explain the discrepancy in the results between Fig. 3A and Fig. 4A? In Fig. 3A, where viral GFP expression is measured at 24 hours post infection, ~100% of cells are infected regardless of NDST1 deletion at a high MOI (i.e., 3). However, in Fig. 4A, where cells were also infected at a high MOI (i.e., 10), but at 24 hours, only ~20% of NDST1-deleted cells were GFP+ compared to 100% in the NDST+ cells.

2. Figure 4B: The data would be better expressed as genome copies (based on comparison to a standard curve). Furthermore, the data are from one experiment. I would suggest repeating the experiment and showing data from three independent experiments, particularly considering the discrepancies observed between the two NDST1-deleted cell lines. In particular, the phenotype in NDST1-deleted cell line #1 appears strange, with viral genomes dropping between 10 and 12 hours.

3. Why were B16/F10 cells chosen for this study? A brief justification of the cell line chose may be helpful for non-experts. Furthermore, BSC40 cells appear to have been used in Figure 4 (as indicated on line 379). Why the change?

4. To confirm that the phenotypes observed are indeed due to the absence of NDST1 (and not clonal selection during cell line construction), I would suggest adding back NDST1 to confirm the effect is reversed by NDST1 expression. Similarly, do the authors see the same phenotype (reduced spread of MYXV, but enhanced spread of VACV) with heparase treatment?

5. The authors nicely explain in the introduction that VACV binds to both heparan sulfate and chondroitin sulfate. Is the reduced effect of NDST1 deletion on VACV compared to MYXV due to its binding to CS? And similarly is the enhanced spread of VACV in the absence of NDST1 facilitated by binding to CS in distant cells? It would be interesting to evaluate the effect of CS removal (i.e., by chondroitinase treatment) in the NDST1 deleted cells, to see whether the VACV phenotype becomes more like the MYXV phenotype.

Minor typographical comments:

1. Line 248: “context” should be “content”

2. Line 315: “typical amounts EEV” should be “typical amounts of EEV”

3. Line 378: “kinetic” should be capitalised to read “Kinetic”

6. PLOS authors have the option to publish the peer review history of their article (what does this mean?). If published, this will include your full peer review and any attached files.

Reviewer #1: No

Reviewer #2: No

---

## [Author Response · Author response to Decision Letter 0]

2 Mar 2020

Response to Reviewer Comments

Editorial Comments:

1) Please ensure that your manuscript meets PLOS ONE's style requirements

We have edited the manuscript for style to conform to Plos One’s requirements (apologies). Note that these changes are not tracked on the marked up file. 

2) PLOS ONE now requires that authors provide the original uncropped and unadjusted images underlying all blot or gel results

Our manuscript only contains 1 figure with western blot data (Fig 1B). The original files for the scanned in gels have now been included as supplemental figures 3 and 4. The scanned in images have been labeled for clarity but not altered in any other way. Note that these supplemental files are not referenced in the manuscript. 

3) Please include your tables as part of your main manuscript and remove the individual files.

Table 1 has now been inserted into the manuscript file as requested. 

Reviewer #1

1) Much of the discussion hinges on the measurement of "plaque size", but how this was determined by the authors is not clearly explained. This is actually quite tricky judging by the images shown in Fig5C or 6A, since there aren't clear margins to the plaques. Is it a measure of the diameter from the outermost of the GFP+ pixels or is it perhaps the area enclosing some % of the GFP+ signal? Some clarity here would be helpful.

We apologize for the lack of clarity. The reviewer is correct in their assumption that plaque size was determined by drawing an area around the outermost ring of GFP+ cells and subsequently determining the area of that region using post image processing (we used ImageJ for this). This is an approach that we have published on previously and we therefore did not include much discussion of this as we simply cited our previous work. We have now removed that citation and instead included a more complete methods section on measuring foci/plaque size. Examples of how we drew the regions around foci are also now included as supplemental Figure S1. 

2) The qPCR data in Fig 4B are odd. Given the discrepant measurements at 10 and 12 hr (in cell line #1 and 2), it's very difficult to come to any conclusions regarding whether DNA synthesis is affected by this mutation. This experiment should be repeated. Ideally one would like to know whether the replication kinetics are any different once the barrier to infection has been overcome (i.e post entry). That means identifying newly infected cells and measuring DNA synthesis in just those cells. Is there any way to determining this? For example, could one measure the rate of growth of DAPI strained virus factories once they are first detected?

We fully agree with the reviewer that our genome quantitation data should have been repeated. We have therefore repeated the previous experiment two additional times as well as also conducted a similar experiment for VACV (which was not previously shown) three separate times. The summation of all three experiments is now shown as new data in Fig 4B. Interestingly, all three replicates for MYXV do show the previously observed decrease in genomes between 10 and 12 hours post infection. While we do not have an explanation for this phenomenon, we hypothesize that it could be due to cellular lysis during this time frame. We would also point out that it occurs in both control and NDST1 deficient cells and is therefore not likely to impact the conclusions of our work. 

3) Lines 251-62. The paragraph surrounding the role of EEV is confusing. The authors lead with a statement that MYX produce much less EEV than VAC, then use methyl cellulose to "inhibit release of EEV", see no effect of methyl cellulose on VAC plating in either cell type, and conclude that EEV play a major role in determining VAC spread. I can't follow the logic, there seems to be a non sequitur or circular reasoning somewhere. As an aside, VAC strain WR doesn't produce much EEV, the effect seen here might be more striking if strain IHD-W was used.

We apologize for the lack of clarity. The phenotype under study was the increased size of VACV plaques seen in NDST1 deficient cells (Fig 5). The data in figure 6 shows that this phenotype goes away in the presence of MC and that under these conditions VACV forms identical sized plaques in both NDST+ and deficient cells. The effect of MC is therefore that it eliminates the previously observed phenotype. The corresponding results section has been edited to hopefully clarify this point. 

4) Lastly, one is still left uncertain whether this mutation is affecting binding alone or binding plus entry via the entry fusion complex. Fig 2 doesn't really help clarify this point and it's only briefly discussed in lines 296 - 300. Have the authors tried to measure a time lag between binding (for example on ice) and uncoating?

We have not directly measured the lag between binding and entry/uncoating. However, our data in Fig 4A measures early gene expression following high MOI infection which is similar to the requested experiments (albeit a slightly more indirect measure). This data clearly shows a delay in GFP expression in NDST1 deficient cells for both VACV and MYXV. We interpreted this as a possible delay in entry in our discussion, however, we could not rule out the possibility that GFP expression in these experiments is virion dose dependent. We have edited the relevant section in the discussion to further emphasize this point. 

5) Abbreviations need to be explained (e.g. NTC)

We apologize for this oversight. We have re-read the manuscript and now provide definitions for all abbreviations used in both the text and the figures upon their first usage. 

6) The green plaque images (e.g. Fig 5A) would be better presented as black on white as in Fig 5C.

We have replaced the images of green foci/plaques in both Fig 5 and Fig 6 with inverted black and white images as requested. 

7) There's an NDST2 I gather, would it still be functional?

These are actually 4 distinct NDST enzymes (NDST1-4). We have now tested expression of all of these enzymes in B16/F10 cells and find expression of NDST1 and 2 (but not 3 and 4). This data is now included as supplemental Figure S2. However, of these four enzymes, NDST1 is absolutely essential for heparin sulfation and this cannot be overcome by the presence of NDST2-4 in the absence of NDST1. Therefore, our KO of NDST1 efficiently prevents all heparin sulfation (Fig 1E) even though NDST2 should still be present in our KO cells (note that we have not directly tested expression of NDST2 in our NDST1 KO cells). A sentence has been added to the results section (line 143) to clarify this. 

8) Can one call NDST1 "essential" as the authors do in the abstract? The k/o cells seem to grow fine.

We apologize for the lack of clarity. NDST1 is not essential for cell growth, however, it is completely essential for GAG sulfation (which is what we were attempting to refer to here). We have edited the referenced section of the abstract to clarify this point. 

9) As an experiment for another day I would be curious to know if virus grown on k/o cells have any subsequent problems infecting normal cells? Presumably the progeny aren't sulfated properly.

This is a very interesting point that we had not initially considered! One would presume that some proteins within the poxviral virion are heparinated and therefore most likely sulfated as well and that this might play a role in poxviral binding or entry (based on the charges involved it might actually be a negative binding regulator). This could probably be examined be looking at the particle:PFU ratios of virions grown in NDST1+ and deficient cells, however, as noted by the reviewer, this likely falls outside the scope of the current manuscript. We’ve added a new paragraph into the discussion section on this issue. 

Reviewer #2

1. Can the authors explain the discrepancy in the results between Fig. 3A and Fig. 4A? In Fig. 3A, where viral GFP expression is measured at 24 hours post infection, ~100% of cells are infected regardless of NDST1 deletion at a high MOI (i.e., 3). However, in Fig. 4A, where cells were also infected at a high MOI (i.e., 10), but at 24 hours, only ~20% of NDST1-deleted cells were GFP+ compared to 100% in the NDST+ cells.

We assume that the review is referring to the VACV results in these figures as the MYXV results appear fairly consistent. Regarding the VACV infection, we completely agree with the reviewer that the results of the two experiments are slightly different. We have attribute these differences to slight experimental variation. Overall infection with VACV in the experiment shown in Fig 4 was inefficient compared to that seen in Fig 3 (Note that the rate of infection at 24 hours in the control NDST1+ cells seen in Fig 4 is actually only 71% which is significantly lower than the 91% seen in Fig 3). While we do not have a concrete explanation for these experimental changes, we would note that poxviral infection is highly dependent on both concentration and the number of freeze thaws a specific stock has undergone (both variables which are difficult to get completely identical over the course of multiple experiments). Regardless of these differences, we do feel that we have included the appropriate control groups in both Fig 3 and Fig 4 and that the results of these groups (while not identical) are within the realm of biological variation and result in fairly consistent conclusions. We therefore feel that the conclusions drawn from these results remain valid. We have included a sentence in the results section (line 202-203) indicating that we believe these differences are due to minor biological variations across experiments. 

2. Figure 4B: The data would be better expressed as genome copies (based on comparison to a standard curve). Furthermore, the data are from one experiment. I would suggest repeating the experiment and showing data from three independent experiments, particularly considering the discrepancies observed between the two NDST1-deleted cell lines. In particular, the phenotype in NDST1-deleted cell line #1 appears strange, with viral genomes dropping between 10 and 12 hours.

Replication of our real time PCR quantitation of viral genomes has been addressed in reviewer #1’s comment #2 above. 

We have previously considered the possibility of generating standard genome curves to present our data as genomes copies. Unfortunately, our experience is that using whole virions for a standard is not accurate due to poor PCR efficiency from intact virus particles (compared to free viral genomes within a cell) and extraction of viral DNA from infected cells yields high levels of cellular genome contamination. We therefore feel that our use of differential CT values represents the most accurate method to display this data. 

3. Why were B16/F10 cells chosen for this study? A brief justification of the cell line chose may be helpful for non-experts. Furthermore, BSC40 cells appear to have been used in Figure 4 (as indicated on line 379). Why the change?

The reviewer raises an excellent point concerning the origination of our study and the reason why B16/F10 cells were chosen. In full disclosure, our lab is primarily interested in oncolytic immunotherapy. In this context, B16/F10 cells represent a standard model for melanoma tumors which represent one of the major targets for virotherapy. The original purpose of removing NDST1 from these cells was to study the role of susceptibility to viral infection during oncolytic treatment (note that a separate manuscript using the NDST1-/- cells in this context is current under preparation). In the process of doing this oncolytic work, we kind of stumbled upon the differential effects of NDST1 loss on MYXV and VACV infection. Due to this background, there is really not a strong scientific rationale for undertaking the basic virology aspect of this work in B16/F10 cells. However, despite the poor initial justification for the cell type used, we do feel that the work remains scientifically strong and that the results obtained are interesting enough to warrant publication. Note that I’m not really sure how to clarify this point coherently in the actual manuscript and so no changes have been made in the text regarding this particular issue. 

The reference to BSC40 cells on line 379 was a typo and has been corrected to refer to NDST1+ or deficient B16/F10 cells. 

4. To confirm that the phenotypes observed are indeed due to the absence of NDST1 (and not clonal selection during cell line construction), I would suggest adding back NDST1 to confirm the effect is reversed by NDST1 expression. Similarly, do the authors see the same phenotype (reduced spread of MYXV, but enhanced spread of VACV) with heparase treatment?

We agree with the reviewer that reconstitution experiments are one method to demonstrate the specificity of CRISPR-based genetic mutations. However, these reconstitution experiments are often not trivial since overexpression comes with numerous caveats. This is one of the major reasons that these types of experiments are rarely performed in any study using CRISPR. In the absence of this type of data, we would argue that both our NDST1 KO cells lines are clonally derived and display identical phenotypes in basically all of our proposed experiments. The chances of this occurring form an off-target, CRISPR mutation in two distinct cell lines is extremely small. Additionally, the phenotypes observed in our studies are highly consistent with previous work on the biology of NDST1 and poxviruses. We therefore feel confident that our results are due to specific loss of NDST1. 

5. The authors nicely explain in the introduction that VACV binds to both heparan sulfate and chondroitin sulfate. Is the reduced effect of NDST1 deletion on VACV compared to MYXV due to its binding to CS? And similarly is the enhanced spread of VACV in the absence of NDST1 facilitated by binding to CS in distant cells? It would be interesting to evaluate the effect of CS removal (i.e., by chondroitinase treatment) in the NDST1 deleted cells, to see whether the VACV phenotype becomes more like the MYXV phenotype.

We agree that the possibility of VACV having a higher binding affinity to CS could play a role in our observed outcomes. Unfortunately, we are not aware of any genetic methodology to specifically eliminate sulfation on CS and genetically eliminating chondroitin itself is likely to have widespread effects which would be difficult to interpret. It would likely be possible to partially overcome this with enzymatic digestion (chondroitinase), however, it’s a fairly large amount of work to do this type of study well and we therefore feel that it is outside the scope of the current manuscript. A sentence has been added to the discussion section concerning the possible role of CS in VACV binding. 

6. Line 248: “context” should be “content”

This typo has been corrected

7. Line 315: “typical amounts EEV” should be “typical amounts of EEV”

This typo has been corrected

8. Line 378: “kinetic” should be capitalized to read “Kinetic”

This typo has been corrected

---

## [Editor Report · Decision Letter 1]

5 Mar 2020

PONE-D-19-34908R1

Reduced cellular binding affinity has profoundly different impacts on the spread of distinct poxviruses.

PLOS ONE

Dear Associate Professor Bartee,

Thank you for submitting your manuscript to PLOS ONE. After careful consideration, we feel that it has merit but does not fully meet PLOS ONE’s publication criteria as it currently stands. Therefore, we invite you to submit a revised version of the manuscript that addresses the points raised during the review process.

Please refer to the Additional Editor Comments below for a full discussion of the issues that need to be addressed.

We would appreciate receiving your revised manuscript by Apr 19 2020 11:59PM. To enhance the reproducibility of your results, we recommend that if applicable you deposit your laboratory protocols in protocols.io, where a protocol can be assigned its own identifier (DOI) such that it can be cited independently in the future. For instructions see: http://journals.plos.org/plosone/s/submission-guidelines#loc-laboratory-protocols

We look forward to receiving your revised manuscript.

Kind regards,

Luis M Schang, MV. Ph.D.

Academic Editor

PLOS ONE

Additional Editor Comments (if provided):

Thank you very much for paying so close attention to all the reviewers and addressing all issues raised by both reviewers. The manuscript has been improved as a result. However, a couple of issues still need some attention, as listed below.

1. Statistics. A few figures show error bars, which are described as SD, and present p values, but are described as presenting the sum of two independent experiments (figure 2B, 3A, 4), or the number of experiments is not presented (figure 5). SD and p values using Student's T test or similar requiere a minimum of 3 independent experiments to test the biological reproducibility of a result. Please clarify what are the error bars and p values. If only two or one biologically independent experiments are presented, then SD and p values cannot be calculated.

2. Wording. "Semi-quantitative", although used in biology, is actually a self-contradictory world: an approach is by definition quantitative or qualitative. In this particular paper, the word does not add much, and it can well be removed (line 178)

3. Clarity to the reader. Although discussed in the discussion section, the presentation of Figure 4 in the results appears contradictory to the results from figure 3. Perhaps you may want to anticipate the likely explanation in the results section, and then present, as you do now, the full discussion in the discussion section? I am of the opinion that it would be helpful to the reader.

4. Lastly, although the discussion about the effects of MC is now far more clear, there is an issue that is not actually discussed. The effect adding MC on VACV plaque size is not by decreasing the size of the plaques in the knockout cells, but rather by increasing the size in the wild-type ones. This may perhaps be a result of experimental variability, but it does not appear to affect the size of MYXV. This unexpected observation deserves some discussion, even it it were just to state that it is an intrinsic experimental variability.

Once again, thank you for paying so close attention to all comments by both reviewers.

---

## [Author Response · Author response to Decision Letter 1]

5 Mar 2020

Additional Editor Comments (if provided):

Thank you very much for paying so close attention to all the reviewers and addressing all issues raised by both reviewers. The manuscript has been improved as a result. However, a couple of issues still need some attention, as listed below.

Please see responses to specific comments below. Note that no changes needed to be made to the figures based on these comments.

1. Statistics. A few figures show error bars, which are described as SD, and present p values, but are described as presenting the sum of two independent experiments (figure 2B, 3A, 4), or the number of experiments is not presented (figure 5). SD and p values using Student's T test or similar require a minimum of 3 independent experiments to test the biological reproducibility of a result. Please clarify what are the error bars and p values. If only two or one biologically independent experiments are presented, then SD and p values cannot be calculated.

We apologize for the confusion. The references in the figure legends to two independent experiments refers to completely separate experiments done on different days. Each experiment was conducted in at least triplicate giving us a total n of 6+. Stats are calculated for the total n which actually gives us relatively solid statistical power. We have amended the figure legends to make this clearer. 

2. Wording. "Semi-quantitative", although used in biology, is actually a self-contradictory word: an approach is by definition quantitative or qualitative. In this particular paper, the word does not add much, and it can well be removed (line 178)

We totally agree with this assessment. We included the semi-quantitative phrase, simply because that is typically how agarose gels are referred to. This has now been replaced with ‘Qualitative” as requested. 

3. Clarity to the reader. Although discussed in the discussion section, the presentation of Figure 4 in the results appears contradictory to the results from figure 3. Perhaps you may want to anticipate the likely explanation in the results section, and then present, as you do now, the full discussion in the discussion section? I am of the opinion that it would be helpful to the reader.

We have now included the following phrase in the results section describing Fig 4. “Indeed, overall VACV infection in this experiment was inefficient compared to other previous infections (Fig 3) which is likely due to minor experimental variation.” Combined with the previously included section about this issue in the discussion we hope that this provides sufficient clarity to the reader concerning the observed results. 

4. Lastly, although the discussion about the effects of MC is now far more clear, there is an issue that is not actually discussed. The effect adding MC on VACV plaque size is not by decreasing the size of the plaques in the knockout cells, but rather by increasing the size in the wild-type ones. This may perhaps be a result of experimental variability, but it does not appear to affect the size of MYXV. This unexpected observation deserves some discussion, even it were just to state that it is an intrinsic experimental variability.

You had to go there. Unlike the data in Figs 3 and 4, this phenotype is highly reproducible suggesting it is not due to experimental variation. We did not discuss it in the previous version of the manuscript since we don’t have a true explanation for it. Our working hypothesis is that this phenotype is the result of MC trapping viral particles close to the site of initial infection thus increasing the local concentration of virus (resulting in more infection). This is actually fairly consistent with our overall model which proposed that NDST1 deficiency causes reduced binding efficacy effectively spreading out VACV infections. While this is only hypothetical, I have added a new paragraph into the discussion regarding this possibility (note that I also had to rearrange some of the surrounding sentences to make the grammer flow properly).

---

## [Editor Report · Decision Letter 2]

6 Mar 2020

PONE-D-19-34908R2

Reduced cellular binding affinity has profoundly different impacts on the spread of distinct poxviruses.

PLOS ONE

Dear Associate Professor Bartee,

Thank you for submitting your manuscript to PLOS ONE. After careful consideration, we feel that it has merit but does not fully meet PLOS ONE’s publication criteria as it currently stands. Therefore, we invite you to submit a revised version of the manuscript that addresses the points raised during the review process.

Please see the "Additional Editor Comments" below.

We would appreciate receiving your revised manuscript by Apr 20 2020 11:59PM. To enhance the reproducibility of your results, we recommend that if applicable you deposit your laboratory protocols in protocols.io, where a protocol can be assigned its own identifier (DOI) such that it can be cited independently in the future. For instructions see: http://journals.plos.org/plosone/s/submission-guidelines#loc-laboratory-protocols

We look forward to receiving your revised manuscript.

Kind regards,

Luis M Schang, MV. Ph.D.

Academic Editor

PLOS ONE

Additional Editor Comments (if provided):

Thank you very much for your prompt response addressing all the previous questions. Regarding the vaccinia plaque size, the discussion added will alleviate the questions that many a reader might have had otherwise. There is one issue that still needs attention, though, regarding statistics. It is not proper to combine replicates of one experiment with biological repeats. Replicates of one experiment test the reproducibility of the measurements, whereas biologically independent experiments test the biological reproducibility of the system. If the experiments were performed twice in triplicates, then, the n is 2, not 6, for the biological reproducibility, and 3, not 6, for the precision of the measurement (which can only be calculated in each experiments separately).

In the case presented in figures 2B, 3A or 4, if they were indeed performed in just two independent experiments, either present the results as average plus/minus ranges or present the results of a single experiment. Unfortunately, it won't be possible to calculate the p values, but this is not critical to the conclusions.

---

## [Author Response · Author response to Decision Letter 2]

13 Mar 2020

Dear Plos Editorial Staff, 

As requested, I am returning our manuscript titled “Reduced cellular binding affinity has profoundly different impacts on the spread of distinct poxviruses” to allow for statistical examination by the Plos One statistical advisory board. 

At issue appears to be our tendency to combine data sets across multiple experiments and subsequently run statistical analysis on the combined data. An example of this has also been uploaded (the example includes the data used in Fig 2B). In this data set we conducted 2 independent experiments with each experiment done in triplicate (note that triplicate here refers to 3 independent wells of cells treated with virus in 3 independent infections with all plating and infections done at the same time). The experiment is analyzed on the flow cytometer which gives us the critical measure for this assay which is reported from the flow cytometer as (Mean : Comp-APC-A = value). The raw data from both experiments is contained in the attached file as well as the analysis leading to the generation of the figure and statistical analysis (note that the data was normalized within each experiment to account for an observed batch effect and hence reported as MFI relative to control). The results indicate strong statistical significance across the total data set which reported in the figure (note that you also get significance across each individual experiment if analyzed separately). 

The Plos One editorial staff objected to this practice claiming that 

“It is not proper to combine replicates of one experiment with biological repeats. Replicates of one experiment test the reproducibility of the measurements, whereas biologically independent experiments test the biological reproducibility of the system. If the experiments were performed twice in triplicates, then, the n is 2, not 6, for the biological reproducibility, and 3, not 6, for the precision of the measurement (which can only be calculated in each experiments separately).

We understand the issue raised by the editor, however, we think our method of presenting all the data would seem to be superior to what many authors use, including authors published in Plos One, which is to present data from one experiment and run stats on the replicates within that experiment. We also don’t feel that there is anything statistically wrong with our method since we do not believe that our samples would be considered ‘linked’ for mathematical purposes (note that I’m not entirely sure what the mathematical definition of a linked sample is, but for the biological purposes of the experiment in question it would likely refer to a single sample which was run on the flow cytometer 3 separate times to yield 3 separate data points – which is not what we did). 

Obviously, I don’t want to publish data which is analyzed correctly and if the Plos One editorial staff feels that the issue raised is valid, we can fairly easily revise the manuscript to address it. I simply want to make sure that this request is both correct and consistent with normal Plos One editorial practices. 

As a final note: we’ve also reexamined all the data contained within our paper and believe that the only place this issue is relevant is in the data contained in Fig 2B. Other instances raised by the editor are examined below. 

Fig 2B: this experiment is done as the editor detailed

Fig 3A: There are no claims of significance made about this data so the statistical analysis can’t really be incorrect (since there are no stats applied). 

Fig 4: Similar to 3A 4A and 4B do not have claims of significance. 4C does claim significance for the myxoma data. For myxoma, this experiment was actually conducted 3 independent times and the stats are therefore run on the averages of all 3 experiments as is being requested by the editor. The vaccinia experiment was only run twice. Since there was not hint of a difference in either experiment we did not run it a third time and no claim of significance was made. The issue here appears to be that the figure legend currently indicates that this experiment were done twice for both viruses which is technically incorrect as one was done twice and one was done 3 times. 

Fig 5 and 6: These figures analyze individual foci measurements. Given the biology of viral foci formation, it’s pretty hard to envision that these would be considered mathematically linked data sets. 

As I mentioned above, we will certainly accept whatever decision is rendered by the editorial staff and can easily revise the manuscript in whatever way is requested. 

Sincerely, 

Eric Bartee

Associate Professor

University of New Mexico Health Science Center

---

## [Decision Letter · Decision Letter 3]

31 Mar 2020

PONE-D-19-34908R3

Reduced cellular binding affinity has profoundly different impacts on the spread of distinct poxviruses.

PLOS ONE

Dear Associate Professor Bartee,

Thank you for submitting your manuscript to PLOS ONE. After careful consideration, we feel that it has merit but does not fully meet PLOS ONE’s publication criteria as it currently stands. Therefore, we invite you to submit a revised version of the manuscript that addresses the points raised during the review process.

Please, see the Additional Editor's comment below

We would appreciate receiving your revised manuscript by May 15 2020 11:59PM. To enhance the reproducibility of your results, we recommend that if applicable you deposit your laboratory protocols in protocols.io, where a protocol can be assigned its own identifier (DOI) such that it can be cited independently in the future. For instructions see: http://journals.plos.org/plosone/s/submission-guidelines#loc-laboratory-protocols

We look forward to receiving your revised manuscript.

Kind regards,

Luis M Schang, MV. Ph.D.

Academic Editor

PLOS ONE

Additional Editor Comments (if provided):

Thank you very much for your prompt reply to the previous review. Please address the question raised by the statistical expert reviewer, whom as you can see consider it appropriate to pool the results.

Reviewers' comments:

Reviewer's Responses to Questions

**Comments to the Author**

1. If the authors have adequately addressed your comments raised in a previous round of review and you feel that this manuscript is now acceptable for publication, you may indicate that here to bypass the “Comments to the Author” section, enter your conflict of interest statement in the “Confidential to Editor” section, and submit your "Accept" recommendation.

Reviewer #3: All comments have been addressed

2. Is the manuscript technically sound, and do the data support the conclusions?

Reviewer #3: Yes

3. Has the statistical analysis been performed appropriately and rigorously? 

Reviewer #3: No

4. Have the authors made all data underlying the findings in their manuscript fully available?

Reviewer #3: Yes

5. Is the manuscript presented in an intelligible fashion and written in standard English?

Reviewer #3: Yes

6. Review Comments to the Author

Reviewer #3: I was asked to look at the statistical issue around the analysis of this basic science paper examining how the loss of GAG sulfation might have impacted on the replication cycles of two model poxviruses, the classic orthopoxvirus VACV as well as the leporipoxvirus myxom (MYXV). I am somewhat surprised that the mechanism for successful binding of these virion families to a host cell remains poorly understood. The magnitude of the effect is large so if proved correct it makes these results potentially useful for developing treatment able to reduce the risk of entry into humans.

The main issue seems to revolve around how the experiment was conducted which led to the key results shown in Figure 2B. The authors uploaded their explanation of the experiment procedure as well as the raw data which were used to draft Figure 2B.

In my opinion, some confusion raised from the term ‘replicate’ as opposed to ‘repeat’. Repeat and replicate measurements are both multiple response measurements taken at the same combination of factor settings; in this case, the settings are the wells of cells treated with viruses in 3 independent infections with all plating and infections done at the same time. However, while repeat measurements are taken during the same experimental run or consecutive runs, replicate measurements are taken during identical but different experimental runs. Repeats are done to increase precision on the same run while replicates are separate experiments all contributing to the overall variability. Form the authors’ explanation it appears that they have used replicates and not repeats so I believe that the Editor’s main comment has been addressed properly.

Therefore, it is correct to treat them as independent and to combine them to obtain the final estimate and use statistical tests for independent statistical units such as the Student t-test. My only suggestion is that because of the small sample size the authors should consider using a non-parametric test instead of the Student T-test which relies on the distribution of the data be approximately normal.

7. PLOS authors have the option to publish the peer review history of their article (what does this mean?). If published, this will include your full peer review and any attached files.

Reviewer #3: No

---

## [Author Response · Author response to Decision Letter 3]

2 Apr 2020

Comment from Reviewer 3:

“…Therefore, it is correct to treat them as independent and to combine them to obtain the final estimate and use statistical tests for independent statistical units such as the Student t-test. My only suggestion is that because of the small sample size the authors should consider using a non-parametric test instead of the Student T-test which relies on the distribution of the data be approximately normal.”

Based on the statistical review of our previous data analysis, it appears that our previously utilized approach of combining multiple experiments is statistically valid. As requested, we have reanalyzed the combined data set using the non-parametric Mann-Whitney test (which also yields significance for the data) and adjusted the figure legend and figure accordingly.

---

## [Editor Report · Decision Letter 4]

6 Apr 2020

Reduced cellular binding affinity has profoundly different impacts on the spread of distinct poxviruses.

PONE-D-19-34908R4

Dear Dr. Bartee,

We are pleased to inform you that your manuscript has been judged scientifically suitable for publication and will be formally accepted for publication once it complies with all outstanding technical requirements.

With kind regards,

Luis M Schang, MV. Ph.D.

Section Editor

PLOS ONE

Additional Editor Comments (optional):

Thank you very much for introducing the minor edition recommended by the statistician. The manuscript is now ready for publication. Thank you very much for considering PLOS One for the manuscript.
---

## [Editor Report · Acceptance letter]

13 Apr 2020

PONE-D-19-34908R4 

Reduced cellular binding affinity has profoundly different impacts on the spread of distinct poxviruses. 

Dear Dr. Bartee:

I am pleased to inform you that your manuscript has been deemed suitable for publication in PLOS ONE. Congratulations! Your manuscript is now with our production department. 

With kind regards,

on behalf of

Dr. Luis M Schang 

Section Editor

PLOS ONE